# Standardised Outcome Reporting for the Nutrition Management of Complex Chronic Disease: A Rapid Review

**DOI:** 10.3390/nu13103388

**Published:** 2021-09-26

**Authors:** Savita A Sandhu, Chloe A Angel, Katrina L Campbell, Ingrid J Hickman, Helen L MacLaughlin

**Affiliations:** 1School of Exercise and Nutrition Sciences, Queensland University of Technology (QUT), Brisbane 4059, Australia; h.maclaughlin@qut.edu.au; 2Healthcare Excellence and Innovation, Metro North Hospital and Health Service, Brisbane 4029, Australia; katrina.campbell@health.qld.gov.au; 3Department of Nutrition and Dietetics, Princess Alexandra Hospital, Brisbane 4102, Australia; i.hickman@uq.edu.au; 4Faculty of Medicine, University of Queensland, Brisbane 4006, Australia; 5Royal Brisbane and Women’s Hospital, Brisbane 4029, Australia

**Keywords:** rapid review, outcomes, outcome measures, nutrition, chronic disease, core outcome set, nutrition intervention

## Abstract

Individuals with coexisting chronic diseases or with complex chronic disease are among the most challenging and costly patients to treat, placing a growing demand on healthcare systems. Recommending effective treatments, including nutrition interventions, relies on standardised outcome reporting from randomised controlled trials (RCTs) to enable data synthesis. This rapid review sought to determine how the scope and consistency of the outcomes reported by RCTs investigating nutrition interventions for the management of complex chronic disease compared to what is recommended by the core outcome sets (COS) for individual disease states. Peer-reviewed RCTs published between January 2010 and July 2020 were systematically sourced from PubMed, CINAHL and Embase, and COS were sourced from the International Consortium for Health Outcomes Measurements (ICHOM) and the Core Outcome Measures in Effectiveness Trials (COMET) database. A total of 45 RCTs (43 studies) and 7 COS were identified. Outcomes were extracted from both the RCTs and COS and were organised using COMET Taxonomy Core Areas. A total of 66 outcomes and 439 outcome measures were reported by the RCTs. The RCTs demonstrated extensive outcome heterogeneity, with only five outcomes (5/66, 8%) being reported with relative consistency (cited by ≥50% of publications). Furthermore, the scope of the outcomes reported by studies was limited, with a notable paucity of patient-reported outcomes. Poor agreement (25%) was observed between the outcomes reported in the RCTs and those recommended by the COS. This review urges greater uptake of the existing COS and the development of a COS for complex chronic disease to be considered so that evidence can be better synthesised regarding effective nutrition interventions.

## 1. Introduction

In clinical trials, outcomes are monitored to ascertain the impact of a prescribed intervention. When conducting research involving nutrition interventions, it is essential that the measured and reported outcomes are clinically relevant and consistent across similar trials. This supports direct comparison across trials and supports the meta-analyses of findings to inform health policy and the development of evidence-based treatments [1,2]. However, heterogeneity in outcome measures is common [3,4], which can prevent timely advancements in healthcare and the integration of evidence into practice.

Core outcome sets (COS) have been developed as one method to harmonise outcomes and to mitigate the repercussions associated with a lack of outcome standardisation [1,2]. COS list the minimum outcomes that a clinical trial should measure and report for a specific condition, ensuring consistency across research [1]. COS aim to the include outcomes that are of the most value to all stakeholders, including patients, practitioners, researchers, and policymakers [5], and are determined using a consensus method. COS are largely compiled by two international agencies: The International Consortium of Health Outcomes Measurement (ICHOM) [6] and the Core Outcome Measures in Effectiveness Trials (COMET) [7]. ICHOM develops COS presented as “Standard Sets”, which are developed using a systematic review to ascertain potential outcome domains, followed by multiple modified Delphi surveys of the stakeholders to select and reach consensus on the outcomes to include [8]. COMET is a database that compiles COS and COS-related literature. COS sourced from COMET have variable development methods, although guidelines do exist [1,9]. Additionally, COMET has published guidelines for COS development and implementation [1] in addition to a taxonomy for outcome classification [10].

However, as COS are largely designed for singular disease states, there is a lack of guidance for implementing COS when testing the efficacy of interventions in populations with multiple co-morbidities. Addressing this limitation is of increasing importance, with multimorbidity now being recognised as one of the greatest challenges facing modern healthcare globally [11,12]. Alongside its rising prevalence, the populations with multimorbidities are amongst the most costly and difficult to treat [13], with impacted individuals frequently experiencing poorer mental health and a lower quality of life, resulting in a higher prevalence of depression and suicide [14,15]. Emerging evidence suggests that the use of existing clinical guidelines and current models of care that focus on single condition management may be failing to provide the necessary support for these complex populations [16].

To address this emerging priority area, the focus of healthcare research and policy is progressively shifting away from individual disease management towards new approaches that simultaneously address multiple disease states through targeting the shared lifestyle-related risk behaviors that underlie chronic disease pathogenesis [16]. Poor diet quality is recognised as a key shared risk factor; therefore, nutrition interventions that focus on changing diet patterns and/or diet quality are of increasing interest for complex chronic disease management [17,18,19,20,21,22]. However, to accurately determine the comparable effectiveness and efficiency of such interventions, it is essential that there is harmonisation within the measuring and reporting of outcomes across clinical trials.

Therefore, this review seeks to evaluate the scope and consistency of the outcomes reported in nutrition intervention RCTs for the management of complex chronic disease (in accordance with the COMET outcome taxonomy) and make comparisons with the outcomes recommended by the COS for single disease states.

## 2. Materials and Methods

This rapid review was undertaken in accordance with the Preferred Reporting Items for Systematic review and Meta-Analysis Protocols guidelines (checklist available in the Appendix A) [23] with the exception of some steps requiring multiple independent assessments. The protocol of this review has been registered through the Centre for Open Science registry (Standardised outcome reporting for the nutrition management of complex chronic disease: a systematic review. Available online: https://osf.io/qdx26 (accessed on 23 September 2021)).

### 2.1. Search Strategy

RCTs and COS available in the English language and that were published between January 2010 and June 2020 inclusive were sought.

Search strategies to identify eligible publications were defined and applied to Embase, PubMed, and the CINAHL databases by two reviewers (CA, SS) with guidance from the authorship team, a liaison librarian, and a postdoctoral researcher. Each strategy was developed from a combination of database specific topic headings and free search terms limited to title and abstract. This included terms pertaining to diet and/or nutrition paired with search terms for each of the four primary chronic conditions (liver disease, cardiovascular diseases, type 2 diabetes, and chronic kidney disease). Terms for the secondary conditions (i.e., metabolic syndrome and features of metabolic syndrome) were not included, as they were required to appear alongside a primary condition. Full search strategies are shown in Appendix A.

For COS, the COMET database was searched by selecting filters for the disease name (population) and publication year. ICHOM was hand searched by selecting relevant disease states.

### 2.2. Inclusion Criteria

#### 2.2.1. Study Population

RCTs were included if the participants met the criteria for complex chronic disease. This was defined as a diagnosis of two or more lifestyle-related chronic diseases (chronic kidney disease (CKD), type 2 diabetes (T2DM), liver diseases, cardiovascular diseases (CVD)), or a diagnosis of one of these plus either a metabolic syndrome (MetS) or a feature of a MetS, including obesity, hypertension, dyslipidaemia, and insulin resistance, as identified in accepted definitions from the International Diabetes Federation (IDF) [24], National Cholesterol Education Program, Adult Treatment Panel III (NCEP ATP III) [25], and American Heart Association/National Heart, Lung, and Blood Institute (AHA/NHLBI) [26]. This allowed for differing definitions of metabolic syndrome to accommodate varying cross-cultural reference ranges for anthropometric measures. For inclusion in this review, the secondary condition (MetS feature) was required to be specified in the study inclusion criteria or present in greater than 90% of the study population. The intention of the included trials must have been to treat both conditions. There were no participant age restrictions.

#### 2.2.2. Types of Interventions

To be eligible for inclusion in this review, interventions were required to include a minimum of ten participants, have a duration greater than 2 weeks, and have a nutritional component that constituted at least 50% of the intervention’s focus. Nutrition intervention was defined as an intentional change to individual nutritional intake for the purpose of disease management. These changes typically involved modifications with respect to the type and/or quantity of food consumed, macronutrient distribution, or energy intake. An intervention was deemed to meet the 50% dietary intervention criteria if diet was required to be the primary focus of the intervention along with supplemental physical activity or other management strategies. RCTs were excluded if the intervention was a single food only (e.g., adding yogurt daily) or if they were supplementation trials (e.g., micronutrient or probiotic supplements).

COS were included if they were designed to be applied to interventions including nutrition intervention. COS that were explicitly relevant to a narrow range of treatment approaches (e.g., surgery, pharmacotherapy) were excluded.

### 2.3. Study Records

#### 2.3.1. Data Management

For the RCTs, the final search results from each database were exported to Endnote reference managing software (Version X9.2, Philadelphia, PA, USA) [27]. Duplicates were removed.

COS from ICHOM [6] were downloaded directly from the ICHOM website, and full-text versions of the COS identified through COMET [7] were accessed through PubMed and Embase. All of the COS were collated using EndNote reference managing software (Version X9.2, Philadelphia, PA, USA) [27].

#### 2.3.2. Selection Process

The RCTs were sorted chronologically in EndNote, and the library was shared between the two lead authors (CA, SS). Articles were assessed for relevance by conducting a screening of the title and abstract then by a full-text screen. A 10% random sample was undertaken in duplicate to ensure the consistency of the screening, and disputes were assessed by a third reviewer (KC) to ascertain the list of included RCTs.

A single reviewer (SS) sourced the COS from ICHOM and conducted a search of COMET. The search details were downloaded from COMET to an Excel spreadsheet, and the articles were assessed for inclusion using reviewer discretion. To be eligible for inclusion, articles were required to identify themselves as a COS within the descriptive title and to have their classification as a COS confirmed as per the listed “study type”. Publishing recency was then used to determine the single most suitable COS for each of the included clinical conditions. In the instance of liver disease/NAFLD (non-alcoholic fatty liver disease), a COS meeting the first criterion was not identified; therefore, the most recent document recommending clinical endpoints was used. Eligibility for inclusion was confirmed by the authorship team.

### 2.4. Data Extraction and Items

For RCTs, study details including title, author/s, publication year, country of publication, study design and aim, sample size, setting, inclusion and exclusion criteria, method of recruitment, intervention type and duration, and conflicts of interest were extracted from the included full-text articles.

Reported data on the outcomes and outcome measures were extracted using a deductive approach according to the COMET outcome taxonomy core areas (death, physiological/clinical, life impact, resource use, and adverse events) [1,2,11,12,16,28]. To note, core areas contain sub-categories called outcome domains, which enable specified outcome classification. Since publication in 2018, the taxonomy has been used in various systematic reviews to evaluate outcome reporting across a range of populations [4,29,30,31]. COS are not obliged to evaluate outcomes from all of the core areas; however, considering a broad range of outcomes may be meaningful for stakeholders [2].

For the purpose of this review, “outcomes” (as defined by the COMET taxonomy), were identified as the stated or implied endpoints of interest being evaluated for a specified intervention, whereas “outcome measures” were defined as the test or instrument used to quantify the change in a specified outcome [1]. For example, an “outcome” of interest may be a change in glycaemic control; however, this may be assessed using various “outcome measures”, including glycated haemoglobin (HbA1c), the quantitative insulin sensitivity check index (QUICKI score), fasting blood glucose (FBG), or other appropriate measures.

Details about the reported outcomes included outcome name, outcome measures, unit of measurement, method of aggregation, and frequency of data collection. These data were grouped under appropriate outcome titles within the respective COMET core area and outcome domain. This taxonomy provided a framework to analyse the number and distribution of outcomes across core areas and outcome domains, thus acting as a structured approach to assess the scope and consistency of outcome reporting [2].

For COS, the that were variables sought were author names, publication year, article type, methods, clinical area, population, treatment approaches, and outcomes requested (outcome name, outcome measures if available, patient population the outcome was to be collected for, unit of measurement, method of aggregation, and data collection frequency).

### 2.5. Data Analysis

For the RCTs, data analysis was performed using descriptive statistics to assess the scope and consistency of outcome reporting using IBM’s Statistical Package for the Social Sciences (SPSS) Version 26. The proportion of publications reporting a minimum of one outcome from each of the COMET core areas and outcome domains was analysed and reported as a percentage of total studies. Detailed analysis of outcome measures was conducted for the three most frequently reported outcomes. Individual analysis of reported outcomes, outcome domains and, core areas was also conducted for each of the four major disease groups to assess the scope and consistency of reporting within groups.

Outcomes reported by COS were first collated and classified according to the COMET taxonomy core areas. Two-way tables were used to highlight agreement between the outcomes reported by the RCTs and the comparable outcomes requested by the COS. When one or more comparable outcomes were evaluated by both an RCT and the relevant COS within a core area, they were noted to be in agreement.

### 2.6. Quality Assessment

The quality of the included RCTs was assessed by two researchers (SS, CA) using the Cochrane Risk of Bias tool (Version 2.0, London, UK) [32]. Each included publication was evaluated as having either a low risk of bias, some concerns, or high concerns for six areas: randomisation process, deviations from the intended interventions (effect of assignment to intervention and effect of adherence to intervention), missing outcome data, outcome measurement, and reporting bias. An overall risk of bias was determined for each RCT using the algorithm [33]. Each researcher assessed 50% of the RCTs, with 10% being assessed in duplicate to ensure congruence. Disagreements were resolved by a joint review of the manuscript to reach consensus.

## 3. Results

A total of 13,395 articles (13,354 RCT publications and 40 COS) were identified. After deduplication, 10,833 records remained and were screened for eligibility through the title and abstract. Full-text copies of 419 studies were screened in their entirety. In total, 52 articles were included in the review, of which 45 were RCT publications (from 43 studies) and 7 of which were COS. This process is illustrated in Figure 1.

### 3.1. Study Characteristics

A full list of included studies is shown in Table 1, including a summary of key study characteristics, including author, year, country of publication, type of nutrition intervention, duration of intervention, and the chronic diseases affecting the population.

#### 3.1.1. RCT Complex Chronic Disease Combinations

Type 2 diabetes mellitus (T2DM) was the most prevalent primary condition, as indicated in 31 of the eligible publications (29 studies; 69% of included RCT publications) [34,38,40,42,43,44,45,46,47,49,50,52,53,54,55,57,58,59,61,63,64,65,67,68,69,70,71,74,77,78]. Non-alcoholic fatty liver disease (NAFLD) and non-alcoholic steatohepatitis (NASH) were the only liver conditions amongst the eligible publications and were reported in eight papers across eight studies (18%) [35,37,39,41,57,60,72,75]. Cardiovascular diseases (CVDs) were the third most common primary condition, occurring in seven publications from seven studies (16%) [36,45,51,56,71,74,76]. Finally, chronic kidney disease (CKD) was included in six publications from separate studies (13%) [42,48,52,62,66,69].

The most common complex disease combination was T2DM and obesity, occurring in 47% of publications (*n* = 21/45, from 19 studies) [34,38,40,43,44,45,46,47,49,50,53,54,57,58,61,63,64,65,68,73,78]. This was followed by NAFLD/NASH and obesity in 13% of publications (*n* = 6/45, from six studies) [35,37,39,41,60,75] and T2DM and hypertension in 7% of studies (*n* = 3/45, from three studies) [52,59,70]. Six (13%) publications, all from distinct studies, had two of the primary chronic disease conditions, including T2DM and CVDs [71,74], T2DM and CKD [42,50,69], and T2DM and NAFLD [67]. The full list of disease combinations is displayed in Table 1.

#### 3.1.2. RCT Interventions

The type and duration of nutrition interventions varied across the publications; however, most of them included an element of energy restriction for weight loss. Twelve article (27%) interventions, all from separate studies, focused on energy restriction with or without macronutrient manipulation (e.g., low carbohydrate, high protein) [34,35,47,54,63,64,68,69,71,73,74,75]. Four publications (9%, all unique studies) were very low-energy diets (VLED), generally including the use of partial or full meal replacements [36,38,43,67]. Six publications (13%, six studies) implemented culturally specific or inspired diets, namely the Mediterranean diet (*n* = 4) [60,62,66,72], Tibetan diet (*n* = 1) [76], and Korean diet (*n* = 1) [52]. Energy-restricted diets high in fiber and coffee but containing limited or no red meat were implemented in four publications (9%) [53,56,65,78]. A full list of interventions for each study is included in Table 1.

#### 3.1.3. COS Included

Seven COS were included from the search of the ICHOM and the COMET database. Two relevant to T2DM were included [4,79], and one COS was included for each of the following: CKD [80], coronary artery disease (CAD) [81], heart failure [82], atrial fibrillation (AF) [83], and NAFLD/NASH [84]. Table 2 describes the COS study details.

### 3.2. RCT Outcome Reporting Scope and Consistency

#### 3.2.1. Outcome Areas

Across the 45 included publications (43 studies), a total of 66 different outcomes were reported, covering all five COMET taxonomy core areas and 25 of the 38 outcome domains. Physiological/clinical core area outcomes were the most frequently reported. All 45 (100%) of the included publications reported at least one outcome from this core area. Conversely, the outcomes from life impact were only cited by 15 publications (15 studies) (33%) [35,40,42,43,45,47,50,54,55,56,57,59,61,70,73]. Outcomes from the adverse events, resource use, and mortality core areas were similarly reported at notably lower frequencies, cited by just 13%, 11%, and 2% of publications, respectively (displayed in Figure 2). Nearly half of all of the publications (47%) only reported outcomes from the physiological/clinical core area.

#### 3.2.2. Outcome Domains

Of the 38 COMET outcome domains, 13 were not captured in any of the included publications. The most frequently cited outcome domains (for which studies reported at least one outcome) were “general outcomes” which were reported by 93% of the publications (*n* = 42); endocrine outcomes, reported by 84% of the publications (*n* = 38); and nutrition and metabolic outcomes alongside vascular outcomes, with both reported by 58% of publications (*n* = 26).

Notably, these are all within the physiological/clinical core area. Information on the reporting of each outcome domain is displayed in Figure 3.

#### 3.2.3. Individual Outcomes and Outcome Measures

A total of 66 different outcomes were identified across the 45 included publications (See Figure 4), with a median of 6 outcomes per RCT publication (IQR = 4). The maximum number of outcomes reported in a single publication was 13, with a minimum of two. A total of 439 different outcome measures were reported across all of the included publications, with a median of nine per publication (IQR = 20). The maximum number of outcome measures in a single study was 89, with a minimum of three.

Across the 66 outcomes, there was limited consistency in terms of outcome reporting, with only five outcomes reported in ≥50% of publications and over half (73%) of all of the outcomes being reported in two publications or fewer. Full information on the frequency of outcome reporting and the number of outcome measures used is contained in Appendix A. 

“Change in body weight/composition” was the most frequently reported outcome, cited in 87% of publications (*n* = 39). Within this outcome, a total of 25 different outcome measures were used to evaluate changes in body weight or composition. The most frequent of these being change in total body mass, BMI, and/or waist circumference (Appendix A). The second most frequently reported outcome was “glycaemic control”, which was identified in 80% of publications (*n* = 36). For this outcome, 22 different outcome measures were utilised, most commonly fasting blood glucose, HbA1C, and fasting insulin (Appendix A). Thirdly, “change in lipid profile” was reported by 64% of the publications (*n* = 29), with 11 different outcome measures (Appendix A). “Dietary intake” was reported in just over half of all of the included publications (53%, *n* = 24). A total of 104 different measures for evaluating dietary intake were identified, with energy intake, macronutrient intake, and fiber intake being the most common (Appendix A). 

### 3.3. COS Outcome Requesting

A total of 69 outcomes were recommended across the seven COS. There was a range of 10 to 18 outcomes and a median number of 12 outcomes per COS. Five of the seven COS evaluated outcomes originated from all five core areas [4,79,81,82,83]. The COS for NASH/NAFLD only requested outcomes from one of the five (20%) core areas (Physiological/Clinical) [84], and the ICHOM’s CKD COS evaluated three of the five (60%) core areas (all except resource use and adverse events) [80].

Using the COMET taxonomy core areas, across the COS 4 outcomes were related to death, 36 to physiological/clinical, 12 to life impact, 7 to resource use, and 10 to adverse events, as illustrated by Table 3.

Figure 4 displays the frequency at which the outcomes within a core area were requested across all COS. Only the two COMET COS consistently detailed the outcome measures to be used to evaluate each outcome [4,84]. ICHOM COS intermittently specified the PRO measure to evaluate the life impact and resource use core area outcomes [79,80,81]. All of the COS rarely specified the outcome measure metric, method of aggregation, or measurement frequency.

### 3.4. Agreement between Outcomes Reported by RCTs and Those Requested by COS

Figure 5 illustrates that 25% agreement was observed between the outcomes reported in the RCT publications and the COS within the COMET taxonomy core areas overall. No RCT publications reported outcomes comparable to those requested by relevant COS across all of the requested core areas. Five of the RCT publications displayed no agreement between their reported outcomes and the COS recommendations [48,50,53,66,76].

The core area with the greatest agreement was physiological/clinical (Figure 6). When a physiological/clinical outcome was requested by a COS, the relevant RCTs reported a comparable outcome 70 out of 82 (85.4%) times. A total of 2 out of 82 (2.4%) opportunities were in agreement for death; 8 of 69 (11.6%) were in agreement for adverse events; and 13 of 74 (17.6%) were in agreement for life impact. There was no (0 out of 74) agreement in the outcomes reported within resource use.

Again, it is important to note that in some cases, more than one COS was applicable to a single RCT. For these, the RCT was evaluated against both COS. Similarly, not all COS evaluated all core areas.

### 3.5. Quality Assessment Results

Based on quality assessment of the included RCTs using the Cochrane Risk of Bias 2.0 tool [32], the majority of studies presented a risk of bias. Of the 45 RCT publications, 7 were deemed to have an overall low risk of bias [43,56,57,64,72,75,76], 17 presented some concerns [34,36,38,40,44,45,46,48,49,51,52,55,58,63,66,67,68], and 21 displayed a high risk of bias [35,37,39,41,42,47,50,53,54,59,60,61,62,65,69,70,71,73,74,77,78]. Appendix A illustrates risk of bias scores for each outcome domain.

## 4. Discussion

To the authors’ knowledge, this is this first rapid review to assess the agreement between the scope and consistency of the outcomes reported in nutrition intervention RCTs including populations with complex chronic disease. The results illustrate extensive heterogeneity in the outcomes and the outcome measures used across similar RCTs. Of the 66 outcomes reported by the 45 RCT publications (43 studies), only five (“body weight/composition”, “glycaemic control”, “diet quality”, “change in lipid profile”, and “blood pressure”) were reported with relative consistency (cited in ≥50% of studies). Similarly, the extracted outcome measures extracted displayed minimal consistency, and extensive variation in the metric values that were used. A narrow scope was observed when reported outcomes were compared to the COMET taxonomy core areas and relevant COS for individual disease states, namely regarding an overwhelming emphasis on physiological/clinical outcomes and an underreporting of non-clinical core areas such as life impact and resource use. While the absence of reporting of mortality core area outcomes likely reflects the duration of the trials, this irregularity contributes to wider inconsistency. Overall, this led to finding little harmony (25%) between the outcomes reported in the RCTs and those recommended by the COS. These limitations inhibit the ability of study findings that can be synthesised and also fail to capture outcomes that may be the most meaningful to patients.

Twenty-four RCT publications (22 studies) had populations with T2DM and hence could be assessed against both included diabetes COS. The RCTs illustrated similar agreement when compared to each COS, as both requested largely analogous outcomes, a notion supported by literature demonstrating the parallels between ICHOM’s diabetes COS and Harman et al.’s T2DM COS [85]. This suggests that COS designed by a consensus methodology are generally consistent with each other. In these circumstances, it may be beneficial for COS developers to refrain from producing additional COS for a specified disease and treatment approach if one is already in existence. The paucity of outcomes assessing life impact and resource use by RCT publications with increasingly complex populations (i.e., RCTs relevant to both diabetes COS and either CKD, NASH/NAFLD, or CAD COS) highlights the urgent need for outcome standardisation to ensure that these areas of patient experience are considered. Most significantly, the RCT publications relevant to the CAD COS hardly reported any the recommended outcomes.

Collectively, this inconsistency in measuring and reporting outcomes undermines the value and usability of the research by preventing the accurate synthesis and comparison of intervention data [3]. Significant heterogeneity in outcome reporting has been repeatedly identified in trials and systematic reviews across a range of disease states, including polycystic kidney disease, haemodialysis, cardiac arrest, neonatal nutrition, and various surgical procedures [2,3,5,86,87,88,89,90]. Notably, the systematic review underlying the T2DM COS sourced from COMET extracted 1444 outcomes from 354 T2DM clinical trials. Outcomes were categorised into 30 of the 38 COMET taxonomy outcome domains, however no singular outcome or outcome domain was reported by all trials [4].

This rapid review identified significant areas for improvement in outcome reporting by clinical trials, specifically the underreporting of the non-clinical areas. These findings are consistent with research across other areas of healthcare. A 2018 systematic review by Sautenet et al. regarding haemodialysis trials noted that over 80% of the outcome measures reported were surrogate or clinical measures, equivalent to the physiological/clinical core area [3]. The frequent omission of relevant patient-reported outcomes (PROs), which largely align with outcomes from the COMET taxonomy core areas of life impact and resource use was also identified, with these being included in just 35% of papers. Similarly, the reporting of PROs has been demonstrated to be as low as 18% for diabetes [91] and 23% for studies on cardiovascular disease [92].

These findings highlight a troubling gap in the scope of current healthcare research with potentially significant implications for clinical practice. The importance of evaluating PROs is well-recognised in the current literature given their value in informing patient-centered care [93]. This is particularly relevant in the context of nutrition interventions due to the social and emotional significance of food [94]. It is critical to consider that adhering to new dietary and/or lifestyle restrictions can be isolating, contributing to loneliness and acting as a barrier to socialisation [95]. Therefore, it is essential that clinical trials monitor patient-reported life impact outcomes such as social and emotional wellbeing to accurately assess the long-term sustainability and appropriateness of such interventions [96]. Failure to do so suggests a narrow scope and may reflect a lack of patient-centered care by overlooking types of outcomes that are potentially valuable to patients [3,97].

Similarly, reporting resource use outcomes in health interventions is strongly recommended to understand the practical considerations associated with an intervention, particularly concerning accessibility and affordability [98]. However, this review found that only five studies reported resource use core area outcomes. In the context of complex chronic disease management, this is especially relevant due to the financial and human resource costs often faced by impacted individuals, their families, and the healthcare system [99].

In saying this, disease-specific PROs are gaining traction for their importance in research [100]. Due to their specificity, disease-specific PROs can call attention to the symptoms that are impacting a patient’s quality of life (QOL) the most, which can help to ascertain treatments that will provide the most value to these populations in both clinical trials and in clinical practice [101]. The literature supports the use of these in multimorbid populations [102]. Of the PROs extracted from all of the COS, only one (“how often someone is admitted to hospital because of their diabetes”, Harman et al.’s T2DM COS [4]) was disease-specific, highlighting the need for the validation and inclusion of these outcomes in broader standardisation efforts.

Analysing extracted outcomes by disease group highlighted the frequent omission of clinically relevant outcomes. Of the 31 publications on individuals with complex T2DM, 16% (5/31) did not report any measure of glycaemic control. This is of note given the pathophysiology of T2DM and the importance of monitoring and optimising blood glucose levels [103]. Concerningly, whilst nutrition interventions were the focus of all of the included RCTs, only 53% of articles recorded dietary intake as an outcome. Similarly, only one COS requested diet-related outcomes [84]. By failing to evaluate individual nutritional intake, it is difficult for studies to establish a link between the intervention (intake) and any observed changes in health status [104], which has significant implications for research quality. While outside the scope of many COS, considering reporting study-specific outcomes in nutrition trials, such as dietary intake, may be important.

Given the complexity of these diet–disease interactions, emerging evidence suggests that the application of metabolomics and metabolomic profiling may be both a reliable and accurate complimentary tool in nutritional trials [105]. Through studying metabolic processes at a cellular level, metabolomics can act as an objective measure of treatment adherence and nutritional intake [106,107] and can also provide insight into the exact mechanisms by which dietary interventions mediate changes in disease outcomes [107]. However, further research and understanding of metabolomics in nutrition is vital in order to ensure the value and accuracy of routine application in clinical trials [108].

Smith et al.’s COS for multimorbidity [28] is the only known COS that can be implemented in clinical trials including participants with more than one disease. This COS was not included in this review, as it did not meet the criteria of being for a singular disease state. Of the final 17 outcomes included in this COS, all of the COMET core areas were recommended except for the physiological/clinical core area. Non-clinical outcomes were featured instead, including patient-reported impacts and behaviors, physical function, healthcare consultation, and health system usage outcomes. The omission of physiological/clinical outcomes enabled the COS to avoid a disease-specific approach, instead emphasisng outcomes outside of this domain to assess the impact of multimorbidity on a patient’s life. In doing so, the recommendations made by this COS challenge current practice.

This review’s findings are relevant to inform clinical research into the nutrition management of complex chronic disease. Given the rising prevalence of multimorbidity [97,109,110,111], creating a new COS for lifestyle-related complex chronic disease would be highly valuable in clinical practice and research to guide health professionals to effectively manage these patients. Developing a standardised set of outcomes to measure and report these outcomes would reduce research waste and stakeholder burden, inclusive of patients and the healthcare system. Alternatively, the feasibility of using the existing Smith et al. multimorbidity COS alongside key clinical outcomes may also be considered. Either way, the use of a robust COS has been strongly indicated as a method to harmonise clinical guidelines, audits, systematic reviews, and quality improvement [85,112,113].

The novelty of this review is a key strength, as it is the first to extract and analyse the outcomes reported in nutrition intervention RCTs and to reconcile these with COS for lifestyle-related chronic diseases. A strong theoretical basis underpinned the data analysis by using the COMET taxonomy to classify the extracted outcomes. With multimorbidity identified as one of the most significant challenges facing healthcare systems globally, this review will inform how health professionals and researchers can effectively research and support these patients in the future, particularly from a dietary perspective, as treating patients with coexisting chronic conditions is becoming the reality of clinical practice. This review also provides valuable insight into how research can more effectively guide the evolution of best-practice healthcare in this area.

A limitation of this rapid review was the single, rather than dual, reviewer methodology for record screening and quality assessment. Although the review largely meets the Cochrane Rapid Review Interim Guidelines, dual review was undertaken for 10% of the sourced articles for this manuscript rather than the recommended 20% [114] due to the significantly high volume of articles that were identified. Significant heterogeneity in rapid review methodologies exists within published works with a high prevalence of single reviewer screening and extraction [115]. An additional limitation was the specificity of the inclusion criteria. Complex chronic disease is a term with varying definitions and does not inherently refer to lifestyle-related diseases. As a result, the findings of this review are limited in scope, even within the context of complex chronic disease.

## 5. Conclusions

The outcomes reported in nutrition intervention RCTs on populations with complex chronic disease are inconsistent, with limited consideration of patient-reported outcomes. There is great discrepancy between the outcomes reported in these trials and those recommended by COS for relevant individual disease states, which undermines the usability of research and acts as a barrier to the effective synthesis of research knowledge for the progression of healthcare practices. Greater implementation of quality COS developed in accordance with COMET guidelines is required to harmonise the scope and consistency of the outcomes reported by RCTs. If a COS for complex chronic disease was to be developed, considering a range of outcomes from all five COMET taxonomy core areas to reflect the needs of all stakeholder groups is recommended. Doing this will enable the identification of effective nutrition interventions to manage the increasing burden of complex chronic disease.

## Figures and Tables

**Figure 1 nutrients-13-03388-f001:**
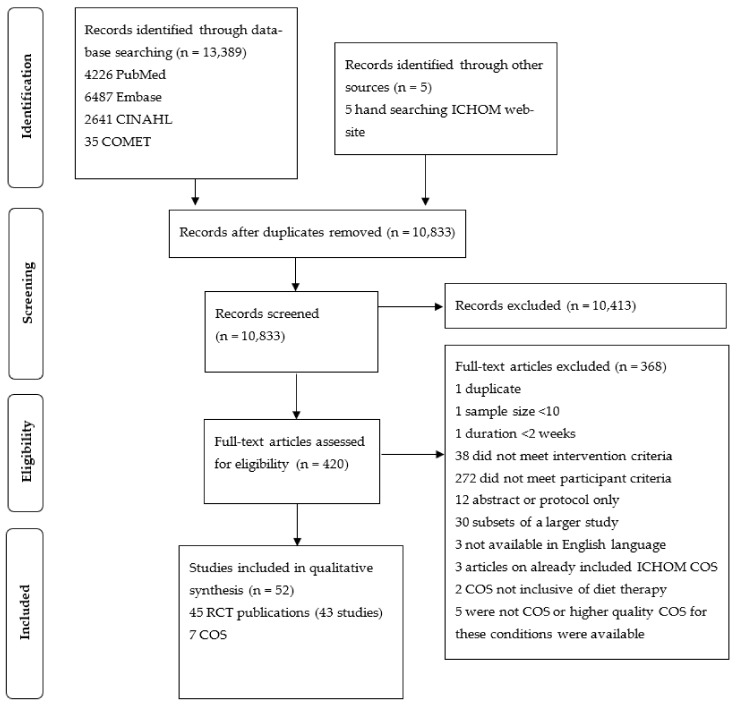
PRISMA flowchart illustrating the study selection process.

**Figure 2 nutrients-13-03388-f002:**
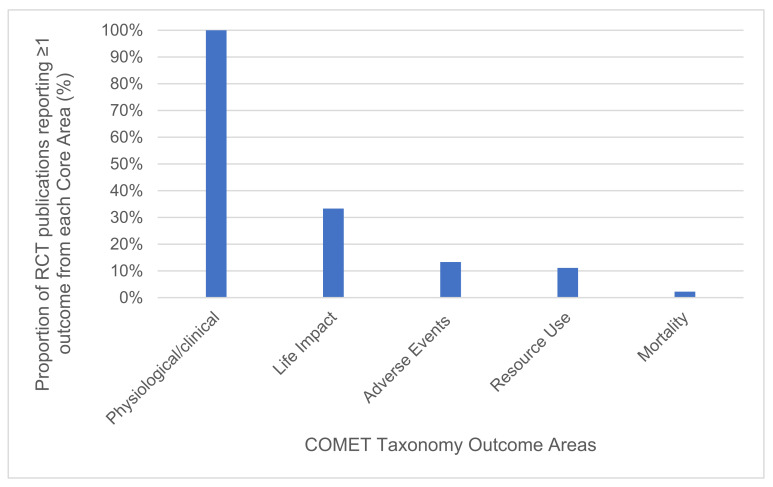
Proportion of RCT publications reporting ≥1 outcome from each COMET outcome core area.

**Figure 3 nutrients-13-03388-f003:**
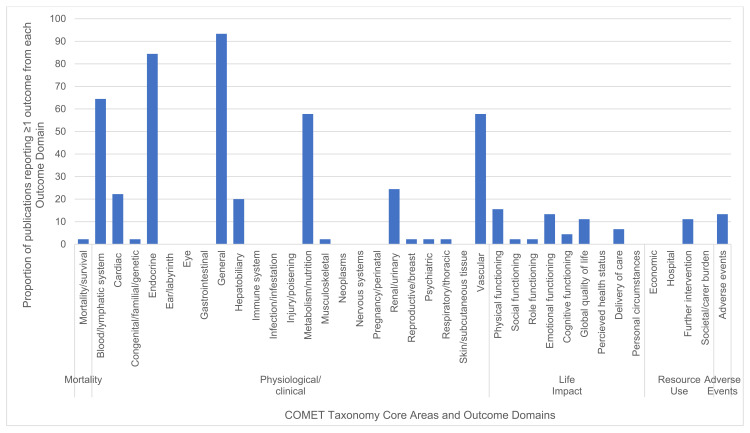
Proportion of studies reporting ≥1 outcome from each outcome domain.

**Figure 4 nutrients-13-03388-f004:**
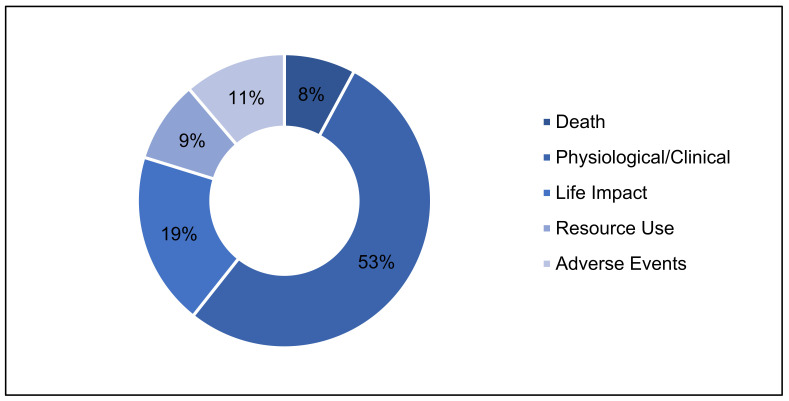
Frequency of outcomes requested by COS, grouped by COMET taxonomy core areas.

**Figure 5 nutrients-13-03388-f005:**
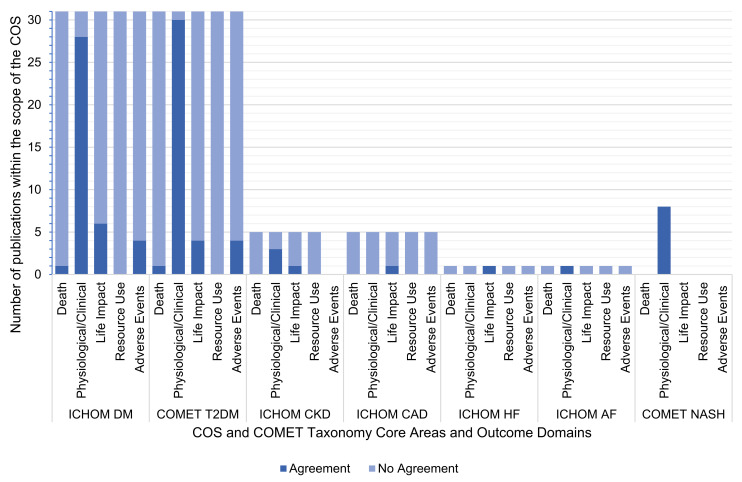
Agreement between outcomes reported by RCTs and their relevant COS, classified according to COMET taxonomy core areas. Note: Outcomes were in agreement when one or more comparable outcome was evaluated by both the RCT and the applicable COS. Some RCTs had more than one COS applicable to them, so they are included under both COS.

**Figure 6 nutrients-13-03388-f006:**
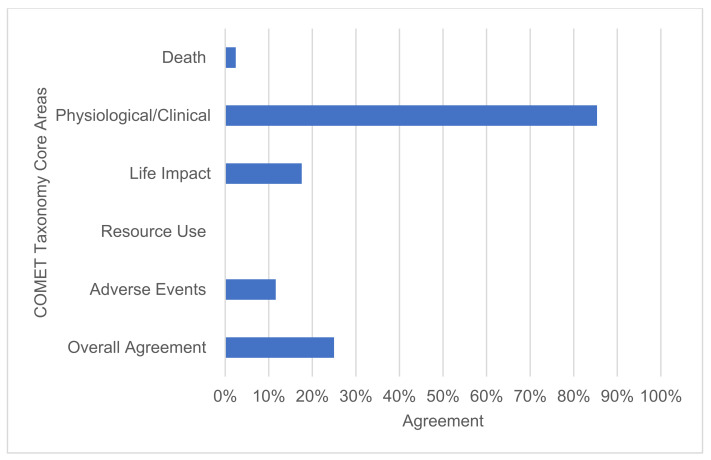
Overall agreement between outcomes reported by RCTs and their relevant COS for each COMET taxonomy core area.

**Table 1 nutrients-13-03388-t001:** Randomised controlled trials for nutritional management of complex chronic disease (arranged alphabetically by author surname).

Author	Year of Publication	Country of Publication	Chronic Disease (1)	Chronic Disease (2)	Dietary Intervention	Duration of Intervention
Abd El-Kader [34]	2016	Saudi Arabia	NASH	Obesity	Energy restricted weight loss diet, standard macronutrient distribution	3 months
Abd El-Kader [35]	2018	Saudi Arabia	T2DM	Obesity	Energy restricted weight loss diet, standard macronutrient distribution	12 weeks
Abed [36]	2013	Australia	AF	Obesity	Very low energy diet with exercise	8-week diet and 15-month follow-up
Aller [37]	2014	Spain	NAFLD	Obesity	Diet enriched with MUFA vs. diet enriched with PUFA	3 months
Brown [38]	2020	United Kingdom	T2DM	Obesity	Low energy diet with liquid total diet replacement formula	3 months
Campos [39]	2012	Brazil	NAFLD	Obesity	Multidisciplinary weight loss program	12 months
Castelnuvo [40]	2011	Italy	T2DM	Obesity	Multidisciplinary weight loss program	4 weeks
Chan [41]	2018	Hong Kong	NAFLD	Obesity	Dietitian led lifestyle modification program	16 weeks, with 52-week maintenance
Ciarambino [42]	2011	Italy	T2DM	CKD	Low protein diet	4 weeks
Corley [43]	2018	New Zealand	T2DM	Obesity	Two consecutive days of VLED (2:5 pattern)	4 weeks
Daniels [44]	2014	United Kingdom	T2DM	Obesity	Diet high in fruit and vegetables	8 weeks + 4-week run-in period
Gallagher [45]	2012	Australia	T2DM/CHD	Obesity	Group multidisciplinary weight loss program	16 weeks
Goldstein [46]	2011	Israel	T2DM	Obesity	Atkins diet	52 weeks
Holland-Carter [47]	2017	United States of America	T2DM	Obesity	Weight watchers weight loss program	52 weeks
Howden [48]	2013	Australia	CKD (St3-4)	MetS component	Multidisciplinary lifestyle modification program	52 weeks
Iqbal [49]	2010	United States	T2DM	Obesity	Low carbohydrate vs. low-fat diet	24 months
Jonsson [50]	2010	Sweden	IHD	Obesity	Paleolithic diet	12 weeks
Jesudason [51]	2013	Australia	T2DM	Obesity/early renal failure	Moderate versus high protein diet	
Jung [52]	2014	South Korea	T2DM	Hypertension	Korean traditional diet	12 weeks
Karusheva [53] *	2018	Germany	T2DM	Obesity	Diet high in cereal fiber and coffee but free of red meat	8 weeks
Khoo [54]	2011	Australia	T2DM	Obesity	Energy restricted diet versus high protein, low carbohydrate	8 weeks with 12-month follow-up
Kim [55]	2011	South Korea	T2DM	Metabolic syndrome	CVD risk reduction program	16 weeks
Kitzman [56]	2016	Germany	Heart failure	Obesity	Energy-restricted diet high in fiber and coffee, but free of red meat	8 weeks
Krebs [57]	2012	New Zealand	T2DM	Obesity	Low fat, high-protein diet	24 months
Luger [58]	2013	Austria	T2DM	Obesity	High protein diet	12 weeks
Lynch [59]	2014	United States of America	T2DM	Hypertension	Dietitian led intensive, group-based diabetes self-management classes	6 months
Martin Alejandre [60]	2019	Spain	NAFLD	Obesity	Energy-restricted diet with high adherence to Mediterranean diet	6 months
Mayer [61]	2014	United States of America	T2DM	Obesity	Low-carbohydrate diet	48 weeks
Mekki [62]	2010	Algeria	CKD	Dyslipidemia/hypertriglyceridemia/hypercholesterolemia	Mediterranean diet	90 days
Mollentze [63]	2019	South Africa	T2DM	Obesity	Energy-restricted, low-fat diet	6 months
Morris [64]	2020	United Kingdom	T2DM	Obesity	Energy-restricted, low-carbohydrate diet	12 weeks
Nowotny [65] *	2015	Germany	T2DM	Obesity	Energy-restricted diet high in fibre and coffee, but low in red meat	8 weeks
Orazio [66]	2011	Australia	Renal transplant	Impaired glucose tolerance	Energy-restricted Mediterranean-style, low GI diet.	2 years
Oshakbayev [67]	2019	Kazakhstan	T2DM	NAFLD	VLED	24 weeks with 24 week follow up
Papakonstantinou [68]	2010	Greece	T2DM	Obesity	Energy-restricted, high-protein low-fat diet	4 weeks for each diet, with 3-week washout
Patil [69]	2013	India	T2DM	Nephropathy	Energy-restricted weight-loss diet	6 months
Paula [70]	2015	Brazil	T2DM	Hypertension	DASH diet	4 weeks
Raygan [71]	2016	Iran	T2DM	CHD	Energy-restricted high- versus low-carbohydrate diet	8 weeks
Ryan [72]	2013	Australia	NAFLD	Metabolic syndrome	Mediterranean diet	2 × 6-week diet periods, plus 6-week washout
Schulte [73]	2020	United States of America	T2DM	Obesity	Weight Watchers diet	12 months
Sixt [74]	2010	Germany	T2DM	CAD	Energy-restricted heart-healthy diet	4 weeks inpatient, plus 5 months outpatient
Utari [75]	2019	Indonesia	NAFLD	Obesity	Energy-restricted, low-fat, low-GI diet	12 weeks
von Haehling [76]	2013	Germany	CAD	Metabolic syndrome	Tibetan diet	12 months
Wing [77]	2013	United States of America	T2DM	Metabolic syndrome	Lifestyle modification program	Median follow-up 9.6 years
Zeigler [78] *	2015	Germany	T2DM	Obesity	Energy-restricted diet high in fiber and coffee but low in red meat	8 weeks

AF: atrial fibrillation, CAD: coronary artery disease, CKD: chronic kidney disease, CVD: cardiovascular disease, DASH: Dietary Approaches to Stop Hypertension, GI: glycaemic index, IHD: ischemic heart diseases, MetS: metabolic syndrome, monounsaturated fat, NAFLD: non-alcoholic fatty liver disease, PUFA: polyunsaturated fat, T2DM: type two diabetes mellitus, VLED: very low energy diet. * publications are from the same study (clinicaltrials.gov NCT01409330).

**Table 2 nutrients-13-03388-t002:** Core outcome set study details.

Author, Date	Date	Database	Clinical Area	Study Type	Methods	Treatment Approaches	Target Population
ICHOM [80]	2017	ICHOM	Chronic kidney disease	COS standard set	Systematic review, multiple modified Delphi surveys, stakeholder consultation, open review	Pre-RRT patients, HD patients, PD patients, transplant patients, conservative care patients	Stage 3a to Stage 5 CKD
ICHOM [79]	2018	ICHOM	Diabetes mellitus, type 1 and type 2	COS standard set		Non-pharmacological therapy, non-insulin-based pharmacological therapy, insulin-based pharmacological therapy	Adults
ICHOM [81]	2015	ICHOM	Coronary artery disease	COS standard set		Lifestyle modification, drug therapy, percutaneous coronary intervention (PCI), coronary artery bypass graft (CABG)	Asymptomatic coronary artery disease, stable angina, acute coronary syndrome (including acute myocardial infarction)
ICHOM [82]	2016	ICHOM	Heart failure	COS standard set		Pharmacotherapy, intensive therapy, rehabilitation	Not further defined
ICHOM [83]	2019	ICHOM	Atrial fibrillation	COS standard set		Management of cardiovascular risk factors, pharmacological management, non-pharmacological management	Not further defined
Harman et al. [4]	2019	COMET	Diabetes mellitus, Type 2	COS for clinical trials or clinical research	Online Delphi survey, face to face consensus meeting	Glucose lowering interventions	Not further defined
Sanyal et al. [84]	2011	COMET	NASH and NAFLD	Classified as COS for clinical trials or clinical research however not explicitly labelled as a COS	Summary of a 2009 workshop on endpoints in NASH	Not stated	Not further defined

**Table 3 nutrients-13-03388-t003:** Comparison of core areas and outcome domains requested by COS.

COMET Core Area	Outcomes	ICHOM DM [79]	COMET T2DM [4]	ICHOM CKD [80]	ICHOM CAD [81]	ICHOM HF [82]	ICHOM AF [83]	COMET NASH/ NAFLD [84]
Death	Non-specific death outcomes	✔□	✔□	✔□	✔□	✔□	✔□	
	Disease specific mortality		✔□					
Physiological/Clinical	Cardiovascular event outcomes		✔□	✔□	✔□		✔□	
	Cerebrovascular outcomes		✔□		✔□		✔□	
	Renal outcomes		✔□	✔□				
	Glycaemic outcomes	✔□	✔□					✔□
	Diabetes events outcomes			✔□	✔□	✔□	✔□	
	Symptom control outcomes			✔□	✔□	✔□	✔□	
	Hepatic outcomes							✔□
	Dietary outcomes							✔□
	Body composition outcomes		✔□					✔□
	Physical activity outcomes							✔□
	Oxidative stress outcomes							✔□
	Lipid profile outcomes							✔□
Life Impact	Physical function outcomes		✔□	✔□	✔□	✔□	✔□	
	Role outcomes						✔□	
	Emotional wellbeing outcomes	✔□			✔□	✔□		
	Quality of life		✔□	✔□	✔□		✔□	
Resource Use	Economic outcomes	✔□						
	Health care use outcomes	✔□	✔□	✔□		✔□	✔□	
	Further intervention				✔□		✔□	
Adverse Events	Adverse outcomes	✔□	✔□			✔□	✔□	
	CV related adverse outcomes					✔□	✔□	

## Data Availability

Not applicable.

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
