# Peer review of "Standardised Outcome Reporting for the Nutrition Management of Complex Chronic Disease: A Rapid Review"

_nutrients, 2021, doi:10.3390/nu13103388_

Round 1
Reviewer 1 Report
I think this paper could be important but the authors need to develop a better reason and data to support their paper. For example, they present a number of papers which are only a small % of the total number of publications and they state obesity, metabolic syndrome ... but don't provide any specific details on the BMI or age or body composition. This paper needs to be rewritten to better present why it is important and a better rational for the proposed changes.
Author Response
Many thanks for your feedback.
- “..they present a number of papers which are only a small percentage of the total number of publications”.
Author Response: We have taken a systematic approach to search strategies and inclusion of studies as described in detail in methods 2.1 and 2.2.
This paper intended to capture nutrition interventions in population groups with complex chronic disease or lifestyle-related morbidity. This review specifically focussed on RCTs employing nutrition interventions (i.e. diet modifications with respect to the type and/or quantity of food consumed, macronutrient distribution or energy intake, not supplements or singular food items) in populations with two or more lifestyle-related chronic diseases (CKD, T2DM, CVDs, liver diseases, MetS or a feature of Mets). To ensure we captured all relevant papers, we designed a broad search strategy with assistance from a librarian and reviewed this large quantity of papers manually. We are confident that all eligible studies have been captured in this rapid review.
- “..they state obesity, metabolic syndrome… but don’t provide any specific details on the BMI or age or body composition…”
Author Response: The purpose of this article is to review the reporting of the outcome. The actual numerical values of the outcomes etc BMI, age, body composition are not relevant but rather whether they were reported in studies or not.
With respect to the inclusion criteria for metabolic syndrome/features of (including obesity, hypertension, dyslipidaemia and insulin resistance), this was created in accordance with accepted definitions from the International Diabetes Federation (IDF) National Cholesterol Education Program, Adult Treatment Panel III (NCEP ATP III) and American Heart Association/National Heart, Lung, and Blood Institute (AHA/NHLBI) (as stated in lines 115-117). However, specific diagnostic cut-off points vary across definitions. Therefore, for the purpose of this review, the accepted cut-off points were sensitive to the criterion used for individual studies. Such that, if the intention of the study was to treat a specified metabolic feature/s and the criteria for diagnosis was clearly stated, the article was eligible for inclusion, regardless of the specific cut-off points used. This was done to allow for variation in definitions of metabolic syndrome and accommodate varying cross-cultural reference ranges for anthropometrical measures.
We analysed the scope and consistency of outcomes reported in Randomised Controlled Trials (RCTs) that evaluated the effectiveness of nutrition interventions in the management of complex chronic disease. Further to this, we analyse how these outcomes compare with those recommended by relevant Core Outcome Sets (COS) and discuss the need and opportunities for optimising outcome standardisation.
The aim of the review is stated on page 2, lines 81-84.
This is the first rapid review to look at outcome reporting in populations with complex chronic disease. The findings demonstrate both extensive heterogeneity and limited scope in outcomes reported by RCTs, with a lack of consideration to patient reported outcomes and failure to report clinically relevant outcomes. These findings are of great significance as the identified limitations undermine the usability of research and act as a barrier to the effective synthesis of research knowledge for the progression of healthcare practices.
We believe that this manuscript is relevant for publication in the Nutrients journal "Nutrition and Diet for Metabolic Health" Special Issue as it features a central focus on optimising the quality and standardisation of outcome reporting of research into nutrition management of chronic diseases of metabolic origin.
Reviewer 2 Report
This is a well-written and well-organized article. there are a few suggestions
- The abstract should be revised to make it concise by reducing the methodology details.
- there are a few redundancies of the contents, need to remove in the revision
- some of the figures generated by the Cochrane Risk of Bias 2.0 tool may be merged or move to a supplementary file.
- in figure 1, "n=419, as full-text article accessed for eligibility", and 368 were excluded, then it should be 51 in remaining not the 52 as mentioned in the paper?
- line number 93-94, "available in the English language" if the initial search was based on selecting only English language, then why in figure 1 it is mentioned that "3 not available in the English language"
Author Response
Many thanks for your feedback, we have taken into account these suggestions and have made the following corrections.
- “The abstract should be revised to make it concise by reducing the methodology details”.
Author Response: The abstract has been revised with removal of extraneous details relating to the methodology. Lines removed include 25-26 regarding details of risk of bias screening and line 27 detailing the COMET Taxonomy Core Areas.
- “There are a few redundancies of the contents, need to remove in the revision”.
Author response: Thank you for noting this. As detailed above, we have removed additional contents from the methods section to make it more concise. We are unsure which sections you are referring to as having extraneous detail. Please let us know if certain sections would benefit from further editing and we will endeavour to make these changes.
- “Some of the figures generated by the Cochrane Risk of Bias 2.0 Tool may be merged or moved to supplementary file”.
Author response: There was only one figure pertaining to the Risk of Bias 2.0 Tool (Figure 7: Quality assessment of RCT using Cochrane Risk of Bias 2.0 Tool). This has now been relocated to supplementary material as recommended.
- “In Figure 1, ‘n=419, as full-text accessed for eligibility’, and 368 were excluded, then it should be 5 remaining not the 52 as mentioned in the paper”.
Author response: We appreciate the noted correction regarding an inconsistency in the Figure 1. PRISMA flowchart. This error has been corrected to reflect the accurate number of included and excluded texts.
- “Line number 93-94 ‘available in the English language’ if the initial search was based on selecting only English language, then why in figure 1 is it mentioned that ‘3 not available in the English language’”.
Author response: This is correct, when conducting the systematic search, the results were selectively filtered by ‘available in the English language’. However, during assessment of the full-texts, three of the articles were found to only have abstracts available in English and therefore were excluded as full-texts could not be reviewed.
Round 2
Reviewer 1 Report
The changes to this paper are appreciated and I think the paper needs to be in the nutrition literature to help improve the standardization of outcomes for future clinical studies on the nutrition management of complex chronic diseases.
This manuscript is a resubmission of an earlier submission. The following is a list of the peer review reports and author responses from that submission.
Round 1
Reviewer 1 Report
It's not very clear what, why , and how you want to accomplish just a lot of words that did not convince me how what you say will improve that current situation but you used only a very small fraction of all the studies in the literature. I think there could be importance in what you are doing but it's not clear. Also, you lost me with the abstract which was too long and not focused - too much unconnected unnecessary information for an abstract.
Author Response
Thank you for taking the time to review and provide feedback on our manuscript. We found your advice constructive and have incorporated many of these suggestions into our revision.
Comment 1: It is not clear what, why and how you want to accomplish, just a lot of words that did not convince me how what you say will improve that current situation. I think there could be importance in what you are doing but it is not clear.
Response 1: Thank you for your comment. The authors have reviewed the entire manuscript, in particularly the abstract and discussion – to make the message clearer and more concise.
Comment 2: You used only a small fraction of all the studies in the literature.
Response 2: This paper intended to capture nutrition interventions in population groups with complex chronic disease or lifestyle-related morbidity. This review specifically focussed on RCTs employing nutrition interventions (i.e. diet modifications with respect to the type and/or quantity of food consumed, macronutrient distribution or energy intake, not supplements or singular food items) in populations with two or more lifestyle-related chronic diseases (CKD, T2DM, CVDs, liver diseases, MetS or a feature of Mets). To ensure we captured all relevant papers, we designed a broad search strategy with assistance from a librarian and reviewed this large quantity of papers manually.
Comment 3: You lost me with the abstract which was too long and not focussed – too much unconnected unnecessary information for an abstract
Response 3: The abstract has been rewritten to be more concise. We have included this below:
Abstract: Individuals with coexisting chronic diseases, or complex chronic disease, are among the costliest and most challenging patients to treat, placing a growing demand on healthcare. Recommending effective treatments, including nutrition interventions, relies on standardised outcome reporting from Randomised Controlled Trials (RCTs) to enable data synthesis. This review sought to determine how the scope and consistency of outcomes reported by RCTs investigating nutrition interventions for the management of complex chronic disease compared to what is recommended by Core Outcome Sets (COS) for individual disease states. Peer-reviewed RCTs published between January 2010 and July 2020 were systematically sourced from PubMed, CINAHL and Embase, and COS were sourced from the International Consortium for Health Outcomes Measurements (ICHOM) and the Core Outcome Measures in Effectiveness Trials (COMET) database. Forty-five RCTs (43 studies) and 7 COS were identified. Cochrane’s Risk of Bias 2.0 tool was used to assess RCT quality. Outcomes were extracted from both RCTs and COS, and organised using COMET Taxonomy Core Areas (Death, Physiological/Clinical, Life Impact, Resource Use, Adverse Events). Sixty-six outcomes and 439 outcome measures were reported by RCTs. RCTs demonstrated extensive outcome heterogeneity, with only five outcomes (5/66, 8%) being reported with relative consistency (cited by ≥50% of publications). Furthermore, the scope of outcomes reported by studies was limited with a notable paucity of patient reported outcomes. Poor agreement (25%) was observed between the outcomes reported in RCTs and those recommended by COS. This review urges greater uptake of existing COS and development of a COS for complex chronic disease to be considered, so that evidence can be better synthesised regarding effective nutrition interventions.
Reviewer 2 Report
I read the manuscript " Standardised outcome reporting for the nutrition management of complex chronic disease: A systematic review" by Angel C et al. with great interest.
It is well organized covering a very important topic in clinical trials of chronic diseases. Chronic diseases multimorbidity is a very common issue in clinical trials and current outcome measures do not capture the effects of nutritional interventions on the patient.
The strength of this review is that it has described the gaps in clinical trials including the partial use of COS as well as their appropriate reporting, and concludes to valuable suggestions with clinical importance.
Two points that this reviewer would like to stress are:
-Regarding the PROs that the authors have identified as underrepresented in current COS it is of utmost importance to be included in the clinical trials design, especially those studying complex chronic diseases.
PROs have been identified to be crucial markers for progression of several diseases, thereby having a clinical meaning (like in psoriasis). Validated disease-specific PROs should be included in COS to reflect the effectiveness of nutritional interventions.
-Nutritional interventions can vary from dietary modifications to supplementations or disease-specific nutritional plans. To establish a causal relationship between the intervention and the COS, we need reliable measures of metabolism that will reflect the cells response to the intervention and their ability to metabolize. Metabolomics is the most reliable tool to study metabolism on a cellular level and it has the potential to provide insights on the links between metabolism and immune response in chronic diseases. In addition, it can monitor patients adherence to intervention. The reviewer suggests that the authors include a small paragraph of this novel technology and the maturity of measuring metabolites as a tool of nutritional assessment and immunometabolism
Author Response
Thank you for taking the time to review and provide feedback on our manuscript. We found your advice constructive and have incorporated these suggestions into our revision. We are pleased to hear that you read it with interested.
Comment 1: Regarding the PROs that the authors have identified as underrepresented in current COS it is of utmost importance to be included in the clinical trials design, especially those studying complex chronic diseases. PROs have been identified to be crucial markers for progression of several diseases, thereby having a clinical meaning (like in psoriasis). Validated disease-specific PROs should be included in COS to reflect the effectiveness of nutritional interventions.
Response 1: Many thanks for identifying this important gap in current COS. We have read into validated disease-specific PROs and can see how these would be valuable to consider in COS for individual disease states. The following text has been incorporated into the discussion, in the context of discussing the paucity of PROs – particularly those from the Life Impact and Resource Use COMET Taxonomy Core Areas.
In saying this, disease-specific PROs are gaining traction for their importance in research [100]. Due to their specificity, disease-specific PROs can call attention to the symptoms impacting a patient’s quality of life (QOL) the most, which can help to ascertain treatments that will provide the most value to these populations in both clinical trials and practice [101]. Literature supports the use of these in multimorbid populations [102]. Of the PROs extracted from all COS, only one (‘how often someone is admitted to hospital because of their diabetes’, Harman et al.’s T2DM COS [4]) was disease-specific; highlighting the need for validation and inclusion of these outcomes in broader standardisation efforts.
Comment 2: Nutritional interventions can vary from dietary modifications to supplementations or disease-specific nutritional plans. To establish a causal relationship between the intervention and the COS, we need reliable measures of metabolism that will reflect the cells response to the intervention and their ability to metabolize. Metabolomics is the most reliable tool to study metabolism on a cellular level and it has the potential to provide insights on the links between metabolism and immune response in chronic diseases. In addition, it can monitor patients adherence to intervention. The reviewer suggests that the authors include a small paragraph of this novel technology and the maturity of measuring metabolites as a tool of nutritional assessment and immunometabolism
Response 2: Thank you for raising this. We have included the following paragraph in the discussion, in the context of discussing the omission of clinically relevant outcomes, including dietary assessment, in nutrition intervention RCTs.
Given the complexity of these diet-disease interactions, emerging evidence suggests that the application of metabolomics and metabolomic profiling may be both a reliable and accurate complimentary tool in nutritional trials [105]. Through studying metabolic processes at a cellular level, metabolomics can act as an objective measure of treatment adherence and nutritional intake [106, 107] and also provide insight into the exact mechanisms by which dietary interventions mediate changes in disease outcomes [107]. However, further research and understanding of metabolomics in nutrition is vital in order to ensure the value and accuracy of routine application in clinical trials [108].
Reviewer 3 Report
- The Abstract has some length and its is very detailed. Please take into account the Instructions for authors. The Abstract should be a total of about 200 words maximum.
- Paragraph 2.3.4 is not necessary. Please remove
- Please provide the version of SPSS used.
- Quality assessment should be performed by two reviewers. Each reviewer should assess all included studies.
The sentence " Each researcher assessed 50% of the RCTs, with 10% in duplicate to ensure congruence" shows wrong methodological approach. - Please provide PRISMA checklist is missing
- Protocol registration for this study is missing. Protocol should be submitted before the research process. In protocol you should define what are you going to investigate, interventions, population, sample size etc. and the reason for this decision
- According to your flowchart, abstract or protocols have been excluded. This is not methodologically correct. You should include abstract if inclusion criteria are met and/or contract with authors for protocols
- Risk of bias 2 is different from ROB 1 and should be performed for each study outcome and not only for each study
Author Response
Thank you for taking the time to review and provide feedback on our manuscript. We found your advice constructive and have incorporated these suggestions into our revision.
Comment 1: The Abstract has some length and it’s is very detailed. Please take into account the Instructions for authors. The Abstract should be a total of about 200 words maximum.
Response 1: Thank you for your comment. The authors have reviewed the abstract to make the message clearer and more concise, as below:
Abstract: Individuals with coexisting chronic diseases, or complex chronic disease, are among the costliest and most challenging patients to treat, placing a growing demand on healthcare. Recommending effective treatments, including nutrition interventions, relies on standardised outcome reporting from Randomised Controlled Trials (RCTs) to enable data synthesis. This review sought to determine how the scope and consistency of outcomes reported by RCTs investigating nutrition interventions for the management of complex chronic disease compared to what is recommended by Core Outcome Sets (COS) for individual disease states. Peer-reviewed RCTs published between January 2010 and July 2020 were systematically sourced from PubMed, CINAHL and Embase, and COS were sourced from the International Consortium for Health Outcomes Measurements (ICHOM) and the Core Outcome Measures in Effectiveness Trials (COMET) database. Forty-five RCTs (43 studies) and 7 COS were identified. Cochrane’s Risk of Bias 2.0 tool was used to assess RCT quality. Outcomes were extracted from both RCTs and COS, and organised using COMET Taxonomy Core Areas (Death, Physiological/Clinical, Life Impact, Resource Use, Adverse Events). Sixty-six outcomes and 439 outcome measures were reported by RCTs. RCTs demonstrated extensive outcome heterogeneity, with only five outcomes (5/66, 8%) being reported with relative consistency (cited by ≥50% of publications). Furthermore, the scope of outcomes reported by studies was limited with a notable paucity of patient reported outcomes. Poor agreement (25%) was observed between the outcomes reported in RCTs and those recommended by COS. This review urges greater uptake of existing COS and development of a COS for complex chronic disease to be considered, so that evidence can be better synthesised regarding effective nutrition interventions.
Comment 2: Paragraph 2.3.4 is not necessary. Please remove
Response 2: Yes, we have removed Paragraph 2.3.4.
Comment 3: Please provide the version of SPSS used.
Response 3: The version of SPSS has been included in Section 2.5.
Comment 4: Quality assessment should be performed by two reviewers. Each reviewer should assess all included studies.
Response 4: This review was undertaken with the intention of reporting on the presence of outcomes, not on treatment effectiveness. Therefore, to balance scientific rigour with the size and scope of the review (which was not treatment effectiveness) we made a number of pragmatic decisions. This resulted in undertaking a systematic search, with each step of a systematic review (title/abstract screening; full text screening; data extraction and quality assessment) having a quality assurance check (i.e. 10% of records undertaken in duplicate (blinded) and congruence check undertaken). This was a pragmatic decision, due to the nature of the review. If this paper was to assess the effectiveness of therapy, then we agree, every step would have been undertaken 100% in blinded duplicate.
Comment 5: The sentence "Each researcher assessed 50% of the RCTs, with 10% in duplicate to ensure congruence" shows wrong methodological approach.
Response 5: As per Response 4 above, as the intention of this review paper is to report the presence of outcomes, not treatment effectiveness, RCT screening was undertaken independently with blinded congruence check of 10% of records (1089 papers for each reviewer). The level of agreement for both cross-checks was greater than 95%.
Comment 6: Please provide PRISMA checklist is missing
Response 6: Thank you for identifying this. The 2020 PRISMA and PRISMA Abstract checklist have been included in the updated Supplementary Materials.
Comment 7: Protocol registration for this study is missing. Protocol should be submitted before the research process. In protocol you should define what are you going to investigate, interventions, population, sample size etc. and the reason for this decision
Response 7: The protocol was registered with COMET prior to preparation and has also been registered with The Centre for Open Science prior to resubmission. This can be found at: https://osf.io/qdx26.
Comment 8: According to your flowchart, abstract or protocols have been excluded. This is not methodologically correct. You should include abstract if inclusion criteria are met and/or contract with authors for protocols
Response 8: As related to Response 4 above, as this review was to report on the scope of outcomes reported in trials, not treatment effectiveness, it was decided that Abstracts, which typically do not include the breadth of outcomes, be excluded. Protocols demonstrate the outcomes ‘intended’ to be reported, the specific aim of our review related to which outcomes are eventually reported in trials. It is for this reason that protocols were also excluded from this review.
Comment 9: Risk of bias 2 is different from ROB 1 and should be performed for each study outcome and not only for each study.
Response 9: Thank you for identifying this. We recognise that for reviews of intervention outcomes, ROB 2 should be performed for each study outcome as bias can vary significantly between these.
The purpose of this review was not to determine the evidence for effectiveness, or GRADE for each outcome assessed, but rather to analyse the scope and consistency of all outcomes reported by included RCTs. For this reason we analysed the outcomes reported by RCTs, rather than the numerical results from specific outcomes. Undertaking quality assessment using ROB 2 for all included RCT outcomes was not considered relevant for the purposes of this review.
Round 2
Reviewer 1 Report
The authors did an excellent job at revising the paper especially for the abstract. I now have a better understand the importance of this paper but think they may have not included all the possible RCTs but they have enough to support greater implementation of quality COS developed in accordance with COMET guidelines.
Author Response
Many thanks for your feedback. We are glad that the importance of the paper is clearer following the revisions suggested.
Through the systematic search and screening process, there were many RCTs excluded on the basis of the population criteria (2 co-existing lifestyle related chronic diseases). Most often, the RCTs considered strictly included participants with a single disease state and/or the co-existing lifestyle-related chronic disease was in an undifferentiated sub-set of the included participants. This reflects the paucity of current research on nutrition interventions for identified multimorbid populations.
Reviewer 2 Report
Accept in present form
Author Response
Thank you. We greatly appreciate your feedback throughout this process.
Reviewer 3 Report
The authors have answered to all comments.
However, the methodological approach is wrong.
The Protocol was submitted on OSF platform on 9th of July
(after the first revision process) and there is only one version.
Please take into consideration the cochrane handbook for any future works.
Author Response
Thank you again for highlighting the importance of methodology. For future works we will take into account the Cochrane handbook and submit the protocol prior to commencing. We did try to lodge the protocol with PROSPERO but due to the recent rule change for submissions it was no longer eligible as we had commenced data extraction.